# Prevalence and predictors of HIV and sexually transmitted infections among vulnerable women engaged in sex work: Findings from the Kyaterekera Project in Southern Uganda

Joshua Kiyingi[1,2], Proscovia Nabunya[1], Ozge Sensoy Bahar[1], Larissa Jennings Mayo-Wilson[3], Yesim Tozan[4‡], Josephine Nabayinda[1‡], Flavia Namuwonge[5‡], Edward Nsubuga[5‡], Samuel Kizito[1‡], Jennifer Nattabi[1‡], Fatuma Nakabuye[5‡], Joseph Kagayi[6‡], Abel Mwebembezi[2‡], Susan S. Witte[7], Fred M. Ssewamala[1]*

1 International Center for Child Health and Development (ICHAD), Brown School, Washington University in St. Louis Brown School, St. Louis, Missouri, United States of America, 2 Reach the Youth Uganda, Kampala, Uganda, 3 Department of Applied Health Science, School of Public Health, Indiana University, Bloomington, Indiana, United States of America, 4 College of Global Public Health, New York University, New York, NY, United States of America, 5 International Center for Child Health and Development (ICHAD), Masaka, Uganda, 6 Rakai Health Sciences Program, Rakai, Uganda, 7 School of Social Work, Columbia University, New York, NY, United States of America

☯ These authors contributed equally to this work.
‡ YT, JN, FN, EN, SK, JN, FN, JK and AM also contributed equally to this work.
* fms1@wustl.edu

**Data Availability Statement:** Data used in this analysis is available upon reasonable request, data access requests can be sent to any of the following

## Abstract

### Introduction

Women engaged in sex work (WESW) have an elevated risk of the human immunodeficiency virus (HIV) and sexually transmitted infections (STI). Estimates are three times higher than the general population. Understanding the predictors of HIV and STI among WESW is crucial in developing more focused HIV and STI prevention interventions among this population. The study examined the prevalence and predictors of HIV and STI among WESW in the Southern part of Uganda.

### Methodology

Baseline data from the Kyaterekera study involving 542 WESW (ages 18–55) recruited from 19 HIV hotspots in the greater Masaka region in Uganda was utilized. HIV and STI prevalence was estimated using blood and vaginal fluid samples bioassay. Hierarchical regression models were used to determine the predictors of HIV and STI among WESW.

### Results

Of the total sample, 41% (n = 220) were found to be HIV positive; and 10.5% (n = 57) tested positive for at least one of the three STI (Neisseria gonorrhoeae, Chlamydia trachomatis and Trichomonas vaginalis) regardless of their HIV status. Older age (b = 0.09, 95%CI = 0.06, 0.13, p≤0.001), lower levels of education (b = -0.79, 95%CI = -1.46, -0.11, p≤0.05),

Associate Deans—at Washington University's Brown School. Provided the conditions outlined below are met, there should not be concern about data sharing. The team is open to data sharing provided the points outlined below, which were part of the study protocol, data sharing plan, and consenting process, are met. Siomari Collazo-Colón, JD Associate Dean for Administration Hillman Hall, Room 254 [o] 314.935.8675 [f] 314.935.8511 Brown School | Washington University in St. Louis [e] scollazo@wustl.edu OR Fred M. Ssewamala, PhD [Also the corresponding author] William E. Gordon Distinguished Professor Associate Dean for Transdisciplinary Faculty Research Professor of Medicine, Washington University School of Medicine Goldfarb, Room 343 Brown School Washington University in St. Louis [o] 314.935.8521 [e] fms1@wustl.edu ♣ A formal research question is specified a priori ♣ Names, affiliations, and roles of any other individuals who will access the shared data; ♣ The deliverable(s)—e.g., manuscript, conference presentation—are specified a priori; ♣ Proper credit and attribution—e.g., authorship, co-authorship, and order—for each deliverable are specified a priori. ♣ A statement indicating an understanding that the data cannot be further shared with any additional individual(s) or parties without the PI's permission; ♣ IRB approval for use of the data (or documentation that IRB has determined the research is exempt) The requestors are expected to handle converting electronic formats (though the research team will consider converting to tab-delimited text format if possible). These conditions were prespecified in our study proposal, study protocol data sharing plan, and consenting and assenting process. Participants enrolled in the study are vulnerable women engaged in sex work, and over 40% of them living with HIV –both highly stigmatized. Thus, to protect this very vulnerable group, we stated in the consent form that only de-identified individual-level data may be shared outside of the research team and only upon completion of the conditions described above.

**Funding:** Kyaterekera study is funded by the National Institute of Mental Health (NIMH) https://www.nimh.nih.gov under award number R01MH116768 (MPIs: FMS and SW). The funders had no role in study design, data collection and analysis, decision to publish, or preparation of the manuscript.

**Competing interests:** The authors have declared that no competing interests exist.

fewer numbers of children in the household (b = -0.18, 95%CI = -0.36, -0.01), p≤0.05), location (i.e., fishing village (b = 0.51, 95%CI = 0.16, 0.85, p≤0.01) or small town (b = -0.60, 95% CI = -0.92, -0.28, p≤0.001)), drug use (b = 0.58, 95%CI = 0.076, 1.08, p≤0.05) and financial self-efficacy (b = 0.05, 95%CI = -0.10, 0.00, p≤0.05), were associated with the risk of HIV infections among WESW. Domestic violence attitudes (b = -0.24, 95%CI = -0.42, -0.07, p≤0.01) and financial distress (b = -0.07, 95%CI = -0.14, -0.004, p≤0.05) were associated with the risk of STI infection among WESW.

## Conclusion

Study findings show a high prevalence of HIV among WESW compared to the general women population. Individual and family level, behavioural and economic factors were associated with increased HIV and STI infection among WESW. Therefore, there is a need for WESW focused HIV and STI risk reduction and economic empowerment interventions to reduce these burdens.

## Introduction

Women engaged in sex work (WESW) in sub-Saharan Africa (SSA) have an elevated HIV burden, estimated at three times higher than the general population [1]. A systematic review found that the estimated HIV burden among WESW in low and middle-income countries was 11.8% [2], compared to 1.8% in high-income countries [3]. In addition, WESW and other key populations are reported to have the highest prevalence of STI in SSA [4]. In Uganda, the burden of HIV among WESW is estimated at 37%, higher than the general female population at 7.6% [5, 6] and the STI prevalence is estimated at 13% for Neisseria gonorrhoeae, 9% Chlamydia trachomatis, 10% Syphilis and 17% for Trichomonas vaginalis [4].

Globally, studies have documented behavioural and structural factors as the leading contributors to the high risk of HIV and STI among WESW [7, 8]. Behavioural factors may include but are not limited to concurrent multiple sexual partners, erratic condom use and type of sexual activity [9–11]. Structural level factors which include economic, social, policy and organizational environment [12], specifically poverty, gender inequality, physical and sexual violence, stigma and discrimination related to commercial sex work, and social exclusion, exacerbate the risk of HIV and STI among WESW [13–15]. WESW are stigmatized, discriminated against and socially marginalized, which significantly increase their risk of HIV infection [7, 10, 16]. Specifically, discrimination and stigmatization affect their abilities to seek medical services like testing, counselling, pre and post-exposure prophylaxis (PrEP and PEP) use, ART treatment and access to condoms from mainstream health care systems [10, 17]. These risks are augmented by drug use and alcohol consumption among WESW [18–20].

In Uganda, sex work is illegal according to the Ugandan Penal Code and the Anti-Pornography Act [21]. Due to the criminality of sex work, few studies have been conducted to determine and document the prevalence of HIV and STI among WESW in Uganda [4, 22, 23]. Therefore, this study aims to contribute to the currently limited data and address knowledge gaps in the prevalence and predictors of HIV and STI among WESW in Southern Uganda. Specifically, the study addresses the following questions: 1) What is the prevalence of HIV and STI among vulnerable women engaged in sex work in Southern Uganda; and 2) What are the critical individual, family-level, behavioural and economic level factors associated with HIV

and STI among WESW? Given the high prevalence of HIV and STI among WESW, study findings may inform the development and implementation of relevant and culturally appropriate programs to engage WESW in HIV/STI prevention and treatment programming in low resource settings, especially those in SSA.

This study is guided by the theory of syndemics [24], which asserts that, within population, the co-occurrence and interaction of multiple adverse conditions produce a much stronger and intense overall health outcome than if each of the conditions were experienced separately [25]. Studies have characterized the HIV pandemic among key populations due to other prevailing problems like mental health disorders, substance use and adverse social conditions that interact with one another and contribute to HIV transmission [25].

## Methodology

### Sample and setting

Baseline data from the Kyaterekera Project *(2018–2023)*, a longitudinal randomized control trial [26] was analyzed. The study aim was to evaluate the efficacy of adding economic empowerment to traditional HIV risk reduction to reduce new incidences of HIV and STI among women engaged in sex work in Uganda (see details in study the protocol [26]). The study recruited 542 WESW, 18 years and above, from 19 HIV hotspots from 7 geopolitical districts of Kyotera, Mpigi, Masaka, Lyantonde, Lwengo, Rakai, and Kalungu. Participants qualified to take part in the study if they were; 1) 18 years and above; 2) reported an episode of unprotected sex in the last 30 days; and 3) reported engaging in transactional sex (exchange of sex for money, good and services) in the last 30 days.

### Participant recruitment

Study participants were identified and recruited from HIV hotspots in South Western Uganda. In each district, the study team engaged community stakeholders working directly with WESW to identify the hotspots where WESW sought economic opportunities [27]. These included towns on significant highways, landing sites and fishing communities with a high prevalence of HIV along Lake Victoria. The study team identified peers or WESW managers (pimps) from these hotspots who later became the site coordinators and helped the study team to mobilize women for recruitment [27]. Details on the study design, intervention and sampling are provided in the study protocol [26].

### Ethical consideration

The study protocol was approved by Uganda Virus Research Institute (UVRI) Ethics Committee (GC/127/18/10/690), the Uganda National Council for Science and Technology (SS4828), the Washington University in St. Louis Institutional Review Board (#201811106) and Columbia University Institutional Review Board (AAAR9804). Participation in the study was voluntary. Written consent from WESW was obtained before participating in the study.

### Data collection

Data were collected using an interviewer-administered survey tool that took 90 minutes to complete. All data collectors were trained in human subject protection and Good Clinical Practice (GCP). All study-related materials, including consent forms and data collection instruments, were translated and back-translated from English to Luganda—the most spoken local language in the study region. The measures were reviewed and approved by language experts from Makerere University, Kampala. The measures used in the study have been used

in previous *Suubi studies* involving the young population and their caregivers, affected by HIV in the study region [28–31] and, other studies which involved WESW [32–34].

## Measures

**HIV and STI sample collection.** HIV and STI prevalence among participants was measured using blood and vaginal fluid samples bioassay. Vaginal swabs were used to collect vaginal fluids to test for STI. Blood samples for HIV testing and viral load were collected from each participant at baseline. Specifically, two HIV-1 enzyme immunoassays (EIAs) were applied to test for HIV-1 serostatus as confirmatory tests according to standard operation. Abbott Determine, Chembio statpak cassette and Abbott Bioline tests were used for HIV testing. Neisseria gonorrhoeae and Chlamydia trachomatis were tested using Nucleic Acid Amplification Tests (NAATs) and culture for Trichomonas vaginalis [26]. NOVA was used to test Neisseria gonorrhoeae, vaxpert for Chlamydia trachomatis and JD Biotech for Trichomonas vaginalis. Experienced nurses and laboratory technicians supervised by the study in-country collaborators at the Rakai Health Sciences Program (RHSP) collected bioassay samples. The process was conducted following the Uganda National Policy Guidelines on HIV Counselling and Testing Program [35]. Samples were transported to the RHSP laboratory for storage and further analysis. Participants with positive HIV results were enrolled in care or referred to a health care unit of their choice by a trained nurse on site. Those who tested negative were enrolled or linked to PrEP services by a PrEP coordinator on site. Participants who were found to have STI received targeted treatment for that STI (i.e., Azithromycin (1 gram) tablets for Neisseria gonorrhoeae and Chlamydia trachomatis, and metronidazole (2 grams) tablets for Trichomonas vaginalis).

**Individual and family-level factors.** Individuals and family-level factors include age (measured in years), marital status (married/in a relationship, single, and other), education level (primary school vs secondary school education), household composition (number of persons and children in the household), attitude towards domestic violence, family cohesion and location (fishing village, small towns, and rural communities). *Family cohesion* was assessed using a Likert scale of five items from the Family Assessment Measure [36] and the Family Environment Scale [37]. The scale evaluates the support that family members give to each other. Participants rated how often each of the five items happened in their family on a 5-point Likert scale (*Never = 1, Sometimes = 2, About half of the time = 3, Most of the time = 4,* and *Always = 5).* High scores indicated high levels of family cohesion (min/max scores = 7–35). *Domestic violence attitude* was measured using the five questions adopted from the COMPASS Program questionnaire [38]. Items assessed whether a husband would be justified to hit or beat his wife if he was annoyed or angered by what the wife does. Participants responded with *no = 0 or yes = 1*, with high scores (max = 5) indicating high levels of domestic violence attitudes.

**Behavioural level factors.** These include sex work stigma, sex debut (the first time a participant exchanged sex for money, goods, drugs, or other services), alcohol use (whether a participant has ever used alcohol or not), number of paying customers in the last 30 days, number of days in a week a participant engaged in sex work, drug use (whether a participant has ever used drugs or not) and condom self-efficacy. Sex Work Stigma index [39] was used to assess *sex work stigma* experienced by the participant. Responses were rated on a 4-point scale (*Strongly disagree = 1, Disagree = 2, Agree = 3, and Strongly agree = 4).* Participants were assessed on their thoughts about other people's reactions once they found out they were engaged in sex work.; high summated scores reflected high levels of sex work stigma with theoretical range of 10–40. *Condom self-efficacy* was measured using Condom Self-Efficacy Scale [40]. Respondents were assessed on their confidence in using condoms with a male sexual

partner, on 8-items with a 3-response point scale (V*ery confident = 1*, *Somewhat confident = 2*, *Not at all confident = 3)*, with a theoretical range of 8–24, higher scores indicated high levels of condom self-efficacy.

**Economic level factors.** Economic level factors include household assets, whether a participant was currently working for pay (in addition to sex work), financial distress, number of income earners in the household and financial self-efficacy. Participants' household assets availability were assessed using a 21-item index; the list included but was not limited to *family level small enterprise*, *a house*, *land*, *gardens*, *means of transportation*, *or means of communication*. A higher index score reflected a more significant number of the participant's household assets. *Financial distress* was assessed using a 5-item Likert scale adapted from the DHS Model A Questionnaire, Uganda Household Survey [41], and Project NOVA [33]. The questions assessed respondent's access to basic needs, such as money for food, housing/accommodation, medical expenses, clothing, and transportation (*Never = 1 and Many times = 4*) with a theoretical range of 5–20. A high score indicated high financial distress. *Financial self-efficacy* was assessed using five items adopted from the Domestic Violence-related Financial Issues (DV-FI) scale [42]. Women were evaluated on their abilities to achieve their specific financial goals. Responses were rated on a 5-point Likert scale, with *Not confident at all = 1*, *Not very confident = 2*, *Somewhat confident = 3*, *Very confident = 4*, and *Extremely confident = 5*. The theoretical range for this scale was 4–20 with higher scores indicating financial self-efficacy.

## Data analysis

Data were analyzed using STATA16.1 (StataCorp, College Station, Texas 77845 USALP, TX, USA). Descriptive analyses were conducted for individual and family level, behavioural level and economic level factors. To examine the prevalence of HIV and STI among WESW, we ran frequencies of HIV and STI test results from the biomarker samples collected at baseline (positive or negative results). To estimate the key individual, family-level, behavioural and economic level factors associated with HIV and STI, three hierarchical regression models were conducted for each of the two outcomes (HIV and STI). Each model controlled for a block of predictors. Model one controlled for individual and family level factors (age, marital status, level of education, household composition, family cohesion, domestic violence attitudes and location), model two controlled for behavioural level factors (sex work stigma, sex debut, number of days involved in sex work, different customers in past 30 days, alcohol use, drug use, condom self-efficacy and STI) and model three controlled for economic level factors (household asset index, financial distress, currently working for pay, number of income earners in the household and financial self-efficacy). Our interest in using the three models determined which combination of the factors better explained the outcome variables. The likelihood ratios for each model were assessed to establish their strength. We considered the statistical significance of all the analyses at p<0.05 and 95% confidence intervals excluding 1.0.

## Inclusivity in global research

Additional information regarding the ethical, cultural, and scientific considerations specific to inclusivity in global research is included in the Supporting Information (S1 File).

## Results

### Descriptive analysis results

Baseline descriptive analysis results are presented in Table 1. Of the total sample (N = 542), the average age of participants was 31.6. About 87% (n = 473) of the participants had primary

**Table 1. Description and characteristics of the population studied.**

| Variable | Total Sample (N = 542) |
|---|---|
| | % (n)/M(SD) |
| *Individual and family-level factors* | |
| Age (Min/Max: 18–55) | 31.6(7.18) |
| Marital Status | |
| Married/ In a relationship | 25.6(139) |
| Single | 13.3(72) |
| Other (divorced, separated, widowed) | 61.1(331) |
| Level of education | |
| Primary school education | 87.3(473) |
| High school education | 12.7(69) |
| Household Composition | |
| Number of people in the household (Min/Max: 1–18) | 3.6(2.18) |
| Number of children in the household (Min/Max: 0–10) | 1.8(1.66) |
| Family cohesion (Min/Max: 7–35) | 24.5(7.0) |
| Domestic violence attitudes (Min/Max: 0–5) | 2.9(1.66) |
| Location | |
| Rural | 22.1(120) |
| Fishing sites | 24.2(131) |
| Small towns | 53.7(291) |
| *Behavioural factors* | |
| Sex work stigma (Min/Max: 10–40) | 29.8(7.77) |
| Sex debut | 24.6(6.19) |
| Days of the week do you engage in sex work (Min/Max: 1–7) | 5.1(1.84) |
| Number of different customers in the past 30 days (Min/ Max: 0–280) | 33.4(47.41) |
| Alcohol use (ever) | 75.3(408) |
| Drug use (ever) | 80.8(438) |
| Condom self-efficacy (Min/Max: 8–24) | 13.0(4.72) |
| *Economic level factors* | |
| Financial distress (Min/Max: 4–20) | 14.4(4.5) |
| Household Asset Index (Min/Max: 0–19) | 5.5(5.2) |
| Currently working for pay (in addition to sex work) | 23.6(128) |
| Number of income earners in the household (Min/Max: 0–4) | 0.9(0.6) |
| Financial self-efficacy (Min/Max: 4–20) | 8.5(4.09) |
| *Outcome variables* | |
| HIV | 41.0(220) |
| STI (women with at least 1 STI) | 10.5(57) |
| STI (by type) | |
| Neisseria gonorrhoeae | 1.3(7) |
| Chlamydia trachomatis | 2.6(14) |
| Trichomonas vaginalis | 7.4(40) |

education, and only 25.6% (n = 139) were married or in a relationship. The average score on domestic violence attitude was 2.9, indicating moderate attitudes. The mean age for participants at sex work debut was 25. Women reported an average of 33 sexual partners in 30 days. About 75.3% (n = 408) of the women had ever used alcohol, and 80.8% (n = 438) had ever used drugs. The average household size was four, with a mean of two children. The average score of financial distress was 13.6 out of 24, which shows a moderate level of financial distress

by participants. The average score of the household asset index was 5.5 out of 21 expected; this indicates low levels of asset ownership. About 23% of the participants worked for pay. Most of the participants in the study were located in small towns (53.7%, n = 291).

## Prevalence of HIV and STI

HIV prevalence among the women was 41% (n = 220), with 34% (n = 75) around fishing communities, 15% (n = 33) in rural communities and 51% (n = 112) in small towns. The overall prevalence of STI was 10.5% (n = 57), with Neisseria gonorrhoeae at 1.3% (n = 7), Chlamydia trachomatis at 2.6% (n = 14) and Trichomonas vaginalis at 7.4% (n = 40). Of the total sample, 4 (0.7%) women tested positive for more than one STI, and 32 (56%) tested positive for both HIV and STI.

## Predictors of testing positive for HIV

Results from hierarchical regression analyses are presented in Table 2. In model 1 (controlling for individual and family level factors), age, education level, location and household composition were associated with a positive HIV test result. Specifically, older women were more likely to test positive for HIV than young women (b = 0.10, 95% CI = 0.21, 0.13, $p \leq 0.001$). Women living near fishing villages (b = 0.52, 95%CI = 0.21, 0.82, $p \leq 0.001$) and those living in small-town communities (b = 0.61, 95%CI = -0.89, -0.32, $p \leq 0.001$) were more likely to test positive for HIV compared to those in rural communities. On the other hand, women with secondary education were less likely to test positive for HIV compared to those with primary education (b = -0.92, 95%CI = -1.57, -0.27, $p \leq 0.01$), and women from households with fewer children were less likely to test positive for HIV (b = -0.18, 95%CI = -0.36, -0.01, $p \leq 0.05$). Model 1 explained 13% of the variance in HIV status (Pseudo $R^2$ = 0.133).

When we controlled for behavioural level factors in mode 2, age (b = 0.09, 95%CI = 0.06, 0.13, $p \leq 0.001$), education level (b = -0.82, 95%CI = -1.49, -0.15, $p \leq 0.01$) and location (fishing community (b = 0.48, 95%CI = 0.16, 0.79, $p \leq 0.01$) and small-town community (b = -0.59, 95%CI = -0.89, -0.29, $p \leq 0.001$) remained significant predictors. In addition, women with high number of different sexual customers (b = 0.04, 95%CI = 0.0001, 0.0090), $p \leq 0.05$%) and those who reported ever having used drugs (b = 0.53, 95%CI = 0.03, 1.02, $p \leq 0.05$) were more likely to test positive for HIV. Model 2 explained 16% of the variance (Pseudo $R^2$ = 0.156).

Similarly, in model 3 where we controlled for economic level factors, age (b = 0.09, 95% CI = 0.06, 0.13, $p \leq 0.001$), education level (b = -0.79, 95%CI = -1.46, -0.11, $p \leq 0.05$), location (fishing community (b = 0.51, 95%CI = 0.16, 0.85, $p \leq 0.01$) and small-town community (b = -0.60, 95% CI = -0.92, -0.28, $p \leq 0.001$)) and ever used drugs (b = 0.58, 95%CI = 0.076, 1.08, $p \leq 0.05$) remained significant predictors for HIV status. In addition, financial self-efficacy was associated with HIV status (b = 0.05, 95%CI = -0.10, 0.00, $p \leq 0.05$). Women with lower scores of financial self-efficacy were more likely to have HIV. Model 3 explained 17% of the variance (Pseudo $R^2$ = 0.168).

## Predictors of testing positive for STI

Table 3 presents results from hierarchical regression analysis estimating the predictors for STI. In model 1 (controlling for individual and family level factors), domestic violence attitudes were associated with STI status. Specifically, women with acceptance attitudes towards domestic violence were less likely to test positive on any of the STI (b = -0.24, 95% CI = -0.404, -0.69, $p \leq 0.001$). Model 1 explained about 8% of the variance in HIV status (Pseudo $R^2$ = 0.0799). In model 2 (controlling for behavioural level factors), domestic violence attitudes remained a significant predictor of STI (b = -0.23, 95%CI = -0.401, -0.057, $p \leq 0.01$). We observed no other significant factors. Model 2 explained 9% of the variance (Pseudo $R^2$ = 0.093). In model 3

**Table 2. Regression analysis (HIV) b(CI).**

| Variable | Model 1: B(95%CI) | Model 2:B(95%CI) | Model 3: B(95%CI) |
|---|---|---|---|
| *Individual and family-level factors* | | | |
| Age | **0.10(0.21, 0.13)**\*\*\* | **0.09(0.06, 0.13)**\*\*\* | **0.09(0.06, 0.13)**\*\*\* |
| Marital status (ref: Other) | | | |
| Married/in relationship | -0.25(-070, 0.19) | -0.24(-0.70, 0.23) | -0.18(-0.66, 0.31) |
| Single | 0.19(-0.40, 0.78) | 0.16(-0.45, 0.77) | 0.12(-0.49, 0.74) |
| Education level (Ref: Primary education) | | | |
| Secondary school education | **-0.92(-1.57, -0.27)**\*\* | **-0.82(-1.49, -0.15)**\*\* | **-0.79(-1.46, -0.11)**\* |
| Household composition | | | |
| Number of people in the household | 0.11(-0.03, 0.25) | 0.11(-0.03, 0.25) | 0.12(-0.036, 0.27) |
| Number of children in the household | **-0.18(-0.36, -0.01)**\* | -0.14(-0.32, 0.04) | -0.15(-0.35, 0.04) |
| Family cohesion | -0.02(-0.05, 0.01) | -0.02(-0.05, 0.01) | -0.02(-0.05, 0.01) |
| Domestic violence attitude | -0.10(-0.22, 0.01) | -0.08(-0.20, 0.04) | -0.08(-0.19, 0.04) |
| Location (Ref: Rural) | | | |
| Fishing communities | **0.52(0.21, 0.82)**\*\*\* | **0.48(0.16, 0.79)**\*\* | **0.51 (0.16, 0.85)**\*\* |
| Small towns | **0.61(-0.89, -0.32)**\*\*\* | **-0.59(-0.89, -0.29)**\*\*\* | **-0.60(-0.92, -0.28)**\*\*\* |
| *Behavioural health factors* | | | |
| Sex work stigma | | -0.00(-0.03, 0.02) | -0.01(-0.03, 0.02) |
| Sex debut | | 0.01(-0.03, 0.05) | 0.01 (-0.03, 0.05) |
| Days engaged in sex work per week | | -0.01(-0.14, 0.11) | -0.00(-0.13, 0.12) |
| Number of different customers | | **0.004(0.000, 0.001)**\* | 0.00(-0.00, 0.01) |
| Drug use (ref: No) | | **0.53(0.03, 1.02)**\* | **0.58 (0.076, 1.08)**\* |
| Alcohol use (Ref: No) | | 0.14(-0.17, 0.81) | 0.32(-0.17, 0.81) |
| Condom self-efficacy | | -0.03(-0.07, 0.02) | -0.02 (-0.07, 0.02) |
| STI | | 0.42(-0.22, 1.05) | 0.48(-0.17, 1.14) |
| *Economic level factors* | | | |
| Financial distress | | | 0.01(-0.03, 0.06) |
| Household Asset Index | | | 0.02(-0.03, 0.07) |
| Currently working for pay | | | -0.47(-0.94, 0.01) |
| Number of income earners in the household | | | -0.08(-0.45, 0.29) |
| Financial self-efficacy | | | **0.05(-0.10, 0.00)**\* |
| LR(df) | 97.44(10)\*\*\* | 17.05(8)\* | 8.23(5) |
| Pseudo R$^2$ | 0.133 | 0.156 | 0.168 |
| P-value | 0.0000 | 0.0000 | 0.0000 |
| AIC | 654.81 | 653.75 | 655.53 |
| BIC | 702.03 | 735.33 | 758.57 |
| N | 541 | 541 | 415 |

\*p≤0.05

\*\*p≤0.01

\*\*\*p≤0.001

(controlling for economic level factors), participants with lower scores of domestic violence attitudes (b = -0.24, 95%CI = -0.42, -0.07, p≤0.01), those living in small-town communities (b = -0.51, 95%CI = -0.94, -0.08, p≤0.05), as well as participants with lower levels of financial distress (b = -0.07, 95%CI = -0.14, -0.004, p≤0.05) were less likely to test positive for any of the STI. Older women were more likely to have STI (b = 0.05, 95%CI = 0.0002, 0.12, p≤0.05). Model 3 explained about 12% of the variance (Pseudo R$^2$ = 0.115).

**Table 3. Regression analysis (STI) b(CI).**

| Variable | Model 1: B(95%CI) | Model 2:B(95%CI) | Model 3: B(95%CI) |
|---|---|---|---|
| *Individual and family-level factors* | | | |
| Age | 0.03(-0.01, 0.70) | 0.04(-0.01, 0.09) | **0.05(0.0002, 0.12)**\* |
| Marital status (ref: Other) | | | |
| Married/in relationship | -0.24(-0.97, 0.48) | -0.28(-1.02, 0.45) | -0.43(-1.19, 0.33) |
| Single | 0.75(-0.05, 1.55) | 0.73(-0.08, 1.77) | 0.74(-0.08, 1.57) |
| Education level (Ref: Primary education) | | | |
| Secondary school education | -1.01(-2.24, 0.21) | -0.96(-2.19, -0.26) | -1.13(-2.38, 0.12) |
| Household composition | | | |
| Number of people in the household | 0.04(-0.12, 0.20) | 0.04(-1.13, 0.21) | 0.05(-0.13, 0.23) |
| Number of children in the household | -0.11(-0.35, 0.12) | -0.12(-0.36, 0.12) | -0.09(-0.35, 0.16) |
| Family cohesion | 0.02(-0.02, 0.06) | -0.02(-0.02, 0.07) | 0.02(-0.02, 0.06) |
| Domestic violence attitude | **-0.23(-0.40, 0.07)**\*\* | **-0.23(-0.40, -0.06)**\*\* | **-0.24(-0.42, -0.07)**\* |
| Location (Ref: Rural) | | | |
| Fishing communities | 0.02(-0.39, 0.43) | 0.07(-0.36, 0.51) | 0.26(-0.23, 0.75) |
| Small towns | -0.33(-0.72, 0.43) | -0.38(-0.78, 0.01) | **-0.51(-0.93, -0.08)**\* |
| *Behavioural health factors* | | | |
| Sex work stigma | | -0.02(-0.05, 0.02) | -0.01(-0.05, 0.03) |
| Sex debut | | -0.03(-0.08, 0.03) | -0.04(-0.10, 0.02) |
| Days engaged in sex work per week | | -0.11(-0.28, 0.07) | -0.11(-0.28, 0.07) |
| Number of different customers | | 0.0002(-0.01, 0.01) | 0.0001(-0.01, 0.01) |
| Drug use (ref: No) | | 0.16(-0.56, 0.89) | 0.14(-0.60, 1.89) |
| Alcohol use (Ref: No) | | 0.26(-0.48, 1.00) | 0.20(-0.55, 0.95) |
| Condom self-efficacy | | 0.3(-0.03, 0.08) | -0.03(-0.03, 0.09) |
| *Economic level factors* | | | |
| Financial distress | | | **-0.07(-0.14, -0.005)**\* |
| Household Asset Index | | | -0.02(-0.09, 0.06) |
| Currently working for pay | | | 0.53(-0.15, 1.23) |
| Number of income earners in the household | | | -0.42(-0.45, 0.11) |
| Financial self-efficacy | | | 0.91(-3.90, 2.06) |
| LR(df) | 28.77(10)\*\*\* | 4.71(7)\* | 7.98(5) |
| Pseudo R$^2$ | 0.079 | 0.093 | 0.115 |
| P-value | 0.0014 | 0.0098 | 0.0073 |
| AIC | 353.24 | 362.53 | 364.55 |
| BIC | 400.47 | 439.81 | 463.300 |
| N | 541 | 541 | 415 |

\*p≤0.05

\*\*p≤0.01

\*\*\*p≤0.001

## Discussion

This study examined the prevalence and predictors of HIV and STI among WESW in Southern Uganda. HIV prevalence was 41%, which is high compared to the prevalence of HIV among the general female population (9.1%) in the region [5, 6]. These results correspond with other studies in the study area [4, 23]. The prevalence of STI was 10.5%, with Neisseria gonorrhoeae at 1.3%, Chlamydia trachomatis at 2.6% and Trichomonas vaginalis at 7.4%. This

is slightly lower than other studies conducted in Uganda (Neisseria gonorrhoeae (13%), Chlamydia trachomatis (9%), Syphilis (10%) and Trichomonas vaginalis (17%) [4, 43].

Consistent with previous studies, individual and household level factors, including age, low education levels, and locality (fishing and small towns), were associated with HIV infection [23, 42, 44–47]. Specifically, in the general population factors such as age, low education level, poverty drug use, alcohol use, and sexual abuse in childhoodincrease the risk of HIV among women [48–50]. Similarly, among WESW, studies in SSA [23, 44] and other parts of the world [44] have documented that older WESW and those with less education are more susceptible to HIV infection. Other studies suggest that older women are often not thought about when designing HIV prevention and reduction programs, and their concerns are rarely addressed by risk reduction interventions [45]. Moreover, low education has been documented as a risk factor for sexual risk behaviours [45] and exposure to HIV and other STI among key populations, including WESW [51, 52].

Our study found that women living near fishing villages and town communities were more likely to receive a positive test result for HIV than those in rural communities. Studies in Uganda have reported that populations in fishing communities and small towns along the highways are at greater risk of HIV infection and excessively contribute to the burden of HIV in Uganda [42, 46, 47]. This is because of the socio-economic dynamics and the lifestyles in the fishing communities and small towns which include uncertainty of making a living [53], fishing seasons (in peak seasons, fishermen pay high money for condomless sex) and multiple sex partners.

Behavioural factors, including the number of different customers and drug use, were associated with HIV infection. This is consistent with previous studies that have documented that WESW with multiple sex partners were at high risk of HIV infection [9–11], and this was augmented by drug use [18–20]. Women under the influence of drugs are less likely to have control over their bodies and negotiating power for safer sex.

Study findings also indicate that low financial self-efficacy was associated with HIV infection among women. The findings are supported by other studies which have documented poverty as one of the primary drivers for women to engage in sex work [13, 26, 53–55]. Women with low financial self-efficacy have less negotiating power with clients regarding safer sex, which puts them at high risk of HIV [32].

Women with more accepting attitudes towards domestic violence were significantly more likely to have STI. Consistent with other studies [56, 57], women with accepting attitudes may not be in a position to negotiate for safer sex with their abusers [57]. Moreover, studies have shown that abused women are more likely to have multiple sexual partners compared to those not abused, which puts them at risk of STI [56].

## Limitations

The study data was self-reported, and there might be some recall bias and social desirability aspects for some variables like the domestic violence attitudes and financial self-efficacy. We could not make any causal inference since cross-sectional data was used in the analysis. Another limitation of the study is potential confounding factors; for instance, we could not explain the association between having fewer children and the high prevalence of HIV among WESW.

Further research is needed to understand the association between household size (children in the household) and HIV prevalence among WESW. Study findings point to the need for WESW-focused efforts that address behavioural health risks to help reduce the burden of HIV and STI among this vulnerable group. Such efforts should emphasize access to user-friendly

information about HIV and STIs risk reduction–taking into account their literacy levels. More economic empowerment programs addressing financial literacy gaps and access to startup capital may be important to help WESW focus on other sources of income, which may lead to better negotiating power with their customers regarding condom use.

## Conclusion

The study findings contribute to the scarce literature of prevalence and predictors of HIV and STI among WESW in SSA, specifically in southern Uganda. Our study findings indicate that the prevalence of HIV among WESW in this region is at 41%, which is much higher compared to the general women population (9.1%) in the study region and the prevalence of STI is 10.5%. Our study findings indicate that older age among WESW, lower levels of education, lower number of children in the household, location (fishing and small towns communities), drug use and intense financial self-efficacy escalate the risk of HIV infection among WESW. Attitudes towards domestic violence and financial distress were associated with the risk of STI infection among WESW. Based on these results, WESW focused HIV/STI prevention programs might help to reduce the infection gap among WESW and the general population.

## Supporting information

**S1 File.**
(DOCX)

## Acknowledgments

We are grateful to women engaged in sex work in Southern Uganda who agreed to participate in the study; this work could not be possible without them. Special thanks to the International Centre for Child Health and Development (ICHAD) at the Masaka Field Office, who coordinated the study in Uganda with study partners, Rakai Health Sciences Program and Reach the Youth Uganda. Lastly, to the research teams at Washington University in St. Louis, Columbia University in New York, Indiana University, and New York University.

## Author Contributions

**Conceptualization:** Joshua Kiyingi, Proscovia Nabunya, Ozge Sensoy Bahar, Susan S. Witte, Fred M. Ssewamala.

**Data curation:** Joshua Kiyingi.

**Formal analysis:** Joshua Kiyingi.

**Funding acquisition:** Fred M. Ssewamala.

**Methodology:** Joshua Kiyingi.

**Project administration:** Flavia Namuwonge, Edward Nsubuga, Fatuma Nakabuye, Joseph Kagayi, Abel Mwebembezi.

**Supervision:** Fred M. Ssewamala.

**Writing – original draft:** Joshua Kiyingi.

**Writing – review & editing:** Proscovia Nabunya, Ozge Sensoy Bahar, Larissa Jennings Mayo-Wilson, Yesim Tozan, Josephine Nabayinda, Samuel Kizito, Jennifer Nattabi, Susan S. Witte.

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
