## [Decision Letter · Decision Letter 0]

19 Dec 2021

PONE-D-21-34621Prevalence and Predictors of HIV and Sexually Transmitted Infections among Vulnerable Women Engaged In Sex Work: Findings from the Kyaterekera Project in Southern UgandaPLOS ONE

Dear Dr. Ssewamala,

Thank you for submitting your manuscript to PLOS ONE. After careful consideration, we feel that it has merit but does not fully meet PLOS ONE’s publication criteria as it currently stands. Therefore, we invite you to submit a revised version of the manuscript that addresses the points raised during the review process.

We kindly ask you to pay extra attention to the  grammar used in the manuscript and to correct accordingly. The method section should also contain a  description of the laboratory method used, as well as the manufacturer with city and country in the event of commercial reagents were used.

We look forward to receiving your revised manuscript.

Kind regards,

Tania Crucitti

Academic Editor

PLOS ONE

Journal Requirements:

We are grateful to women engaged in sex work in Southern Uganda who agreed to participate in the study; this work could not be possible without them. Special thanks to the team at the International Centre for Child Health and Development (ICHAD), who coordinated the study in Uganda with study partners, Rakai Health Sciences Program and Reach the Youth Uganda. Lastly, to the research teams at Washington University in St. Louis, Columbia University in New York, Indiana University, and New York University.   

Kyaterekera study is funded by the National Institute of Mental Health (NIMH) under award number R01MH116768 (MPIs: Fred Ssewamala, PhD & Susan Witte, PhD). NIMH was not involved in the study design, data collection, analysis, findings interpretation and manuscript preparation. The content in this article does not reflect the views of NIMH or the National Institutes of Health.

We note that you have provided funding information. However, funding information should not appear in the Acknowledgments section or other areas of your manuscript. We will only publish funding information present in the Funding Statement section of the online submission form. 

Kyaterekera study is funded by the National Institute of Mental Health (NIMH) https://www.nimh.nih.gov

under award number R01MH116768 (MPIs: FMS and SW). The funders had no role in study design, data collection and analysis, decision to publish, or preparation of the manuscript.

Reviewers' comments:

Reviewer's Responses to Questions

**Comments to the Author**

1. Is the manuscript technically sound, and do the data support the conclusions?

Reviewer #1: Partly

Reviewer #2: Yes

2. Has the statistical analysis been performed appropriately and rigorously? 

Reviewer #1: Yes

Reviewer #2: Yes

3. Have the authors made all data underlying the findings in their manuscript fully available?

Reviewer #1: No

Reviewer #2: Yes

4. Is the manuscript presented in an intelligible fashion and written in standard English?

Reviewer #1: No

Reviewer #2: Yes

5. Review Comments to the Author

Reviewer #1: thank you for the opportunity to review this paper. It provides interesting insights into a group that is of particular interest for HIV prevention.

General comments:

1)Please run the paper through a program like grammarly as there are a number of grammatical errors

2)Write out full name of bacterial STI’s

3)Why not include syphilis testing if rates in WESW are so high

Intro/Methods/results/Conclusions

4)STI prevalence instead of STIs prevalence

5) What is the general population STI prevalence

6) STI prevalence is estimated at 13% for gonorrhea, 9% chlamydia, 10% syphilis and 17% for trichomonas [4]. Is this for genital STI’s ?

7) Please give the names of the tests you used for HIV and STI

8) Participants who were found to have STIs received single dose of treatment – rather say received targeted treatment for that STI.

9)Age usually reported as a median

10) STI levels seem lower than expected

11) Report results consistently xx% (n= yy)

12) Women with more than one STI were 0.7%, - this does not make sense. Rather say xx women (0.7%) tested positive for more than one bacterial STI.

13) STIs prevalence is estimated at 13% for gonorrhea, 9% chlamydia, 10% syphilis and 17% for trichomonas [4]. From intro.

Same sentence in discussion says syphilis is 17% with same citation

14) Line 294 is obscured

15)The discussion makes some statements that seem to be more author opinion than actual study conclusions. The authors should refrain from hypothesizing reasons for associations unless these can be backed up by literature. For example saying that older women fear stigma more than young women could be completely incorrect. This is an author hypothesis and not a study result. You could say that xxx study showed this to be true and this could explain the association you found in your study. In the same way the authors should be very careful of suggesting interventions based on associations. It would be preferable to say “we noted an association between xxx and yyy and further research is needed to understand this association.

16) The discussion should also compare predictors of HIV in the general female population with WESW. It is understood that HIV prevalence is higher but would be interesting to understand if predictors are different.

17) Line 298 – Prevalence of HIV tends to increase with increasing age. The trend of increasing prevalence in HIV in WESW seems to be the same as for the general population and is an epidemiological phenomenon with a disease like HIV that is incurable.

18) Please provide evidence from your study supporting these statements:

This may be attributed to stigma or fear of accessing preventive methods (e.g., buying condoms, PrEP and PEP services) by older women.

19) Line 306 – please provide evidence for the following statement

Women from households with more children may be less likely to engage in high risk behaviours for fear that children will notice. They may also be more likely to think about the continuity of providing for their children if they are the primary caregivers, therefore taking extra precautions/preventive measures about their lives compared to those with less children

20)Line 306 should be “fewer” children. Do you have any evidence to back up your theories of why fewer children would be a risk factor for HIV. Could this be confounding?

21) Line 330 needs to be rewritten

22) Line 333 – Do you mean accepting attitudes rather than approving attitudes?

23) Line 342 is incomplete

24) A major limitation of this study is potential confounding. The association between low levels of education and HIV may actually be caused by poverty. Educational programs may not be helpful in this context but poverty alleviation programs may work. The authors should be careful about speculating on interventions based on associations found.

Reviewer #2: This study is important for understanding and acting on the health conditions of women sex workers in Uganda. It is guided by syndromic theory, which states that, within the population, the co-occurrence and interaction of multiple adverse conditions produce stronger and more intense overall health outcomes than if each condition were experienced separately.

It shows a major interest in taking care of these disadvantaged women and allowing them to enter an HIV prevention program by administering PrEP in Africa.

The article is well written.

General comments:

Although not the focus of this study, the authors did not explore the prevalence of HPV, which could have been an important opportunity for cervical cancer screening and prevention.

Were patients with a bacterial STI symptomatic?

Were patients with a detected STI treated and with what treatment?

Minor comments

Line 139: could the author precise the apparatus and kit use for NAAT

Line 222: The average age of participants was 31.4% in the text and 31.6% in the table 1

The status married was 25.7 in the text and 25.6 in the table 1

Table 1:

- Titre “Sample characteristics” change for “Description and characteristics of population studied “

- Legends of the Table: “Total Sample (N=542) % (n)” could be precise to understand that the author gives sometimes the %, or the number or the average score.

- What means the number 7.18?

- The total of % is not correct, change 87.7%(473) to 87.3%

Line 292: While the percentage of HIV-positive people is very high, the author observed a low level of bacterial STIs. How can this be explained in this high risk population? Is the NAAT test used for CT/NG/TV screening sufficiently sensitive?

Line 293: Trichomonas without capitalization

6. PLOS authors have the option to publish the peer review history of their article (what does this mean?). If published, this will include your full peer review and any attached files.

Reviewer #1: No

Reviewer #2: No

---

## [Author Response · Author response to Decision Letter 0]

15 Feb 2022

Prevalence and Predictors of HIV and Sexually Transmitted Infections among Vulnerable Women Engaged in Sex Work: Findings from the Kyaterekera Project in Southern Uganda

Response to Reviewers’ Comments

Academic Editor 

Response: We have followed the PLOS ONE’s manuscript required style. 

Response: We have completed the questionnaire and it is attached. A subsection under the methods section has been added to reference the questionnaire. 

We are grateful to women engaged in sex work in Southern Uganda who agreed to participate in the study; this work could not be possible without them. Special thanks to the team at the International Centre for Child Health and Development (ICHAD), who coordinated the study in Uganda with study partners, Rakai Health Sciences Program and Reach the Youth Uganda. Lastly, to the research teams at Washington University in St. Louis, Columbia University in New York, Indiana University, and New York University. Kyaterekera study is funded by the National Institute of Mental Health (NIMH) under award number R01MH116768 (MPIs: Fred Ssewamala, PhD & Susan Witte, PhD). NIMH was not involved in the study design, data collection, analysis, findings interpretation and manuscript preparation. The content in this article does not reflect the views of NIMH or the National Institutes of Health.

We note that you have provided funding information. However, funding information should not appear in the Acknowledgments section or other areas of your manuscript. We will only publish funding information present in the Funding Statement section of the online submission form. 

Kyaterekera study is funded by the National Institute of Mental Health (NIMH) https://www.nimh.nih.gov under award number R01MH116768 (MPIs: FMS and SW). The funders had no role in study design, data collection and analysis, decision to publish, or preparation of the manuscript.

Response: We have removed the funding statement from the manuscript. The version submitted is correct. We are not amending the statement.

Response: We acknowledge this concern. Please see below:

The study is still ongoing. Upon study completion, data access requests can be sent to any of the following Associate Deans—at Washington University’s Brown School. Provided the conditions outlined below are met, there should not be concern about data sharing. The team is open to data sharing provided the points outlined below, which were part of the study protocol, data sharing plan, and consenting process, are met.

Siomari Collazo-Colón, JD 

Associate Dean for Administration

Hillman Hall, Room 254

[o] 314.935.8675 [f] 314.935.8511 

Brown School | Washington University in St. Louis

[e] scollazo@wustl.edu

OR

Fred M. Ssewamala, PhD [Also the corresponding author]

William E. Gordon Distinguished Professor

Associate Dean for Transdisciplinary Faculty Research

Professor of Medicine, Washington University School of Medicine

Goldfarb, Room 343

Brown School Washington University in St. Louis

[o] 314.935.8521 [e] fms1@wustl.edu

A formal research question is specified a priori

Names, affiliations, and roles of any other individuals who will access the shared data;

The deliverable(s)—e.g., manuscript, conference presentation—are specified a priori;

Proper credit and attribution—e.g., authorship, co-authorship, and order—for each deliverable are specified a priori.

A statement indicating an understanding that the data cannot be further shared with any additional individual(s) or parties without the PI’s permission;

IRB approval for use of the data (or documentation that IRB has determined the research is exempt)

The requestors are expected to handle converting electronic formats (though the research team will consider converting to tab-delimited text format if possible).

These conditions were prespecified in our study proposal, study protocol data sharing plan, and consenting and assenting process. Participants enrolled in the study are vulnerable women engaged in sex work, and over 40% of them living with HIV –both highly stigmatized. Thus, to protect this very vulnerable group, we stated in the consent form that only de-identified individual-level data may be shared outside of the research team and only upon completion of the conditions described above.

Response: The reference list has been reviewed and it is complete and correct.

 

REVIEWERS’ COMMENTS

Reviewer #1: General comments:

1. Please run the paper through a program like grammarly as there are a number of grammatical errors

Response: The manuscript has been run through grammarly and all grammatical errors have been corrected. 

2. Write out full name of bacterial STI’s

Response: Full names have been provided as follows: Chlamydia trachomatis, Neisseria gonorrhoeae, and Trichomonas vaginalis

3. Why not include syphilis testing if rates in WESW are so high?

Response: We acknowledge the reviewer’s concern. However, due to budgetary implications, we were only able to test for Gonorrhea, Trichomonas, Chlamydia and HIV. Participants who expressed concern that they might be exposed to syphilis were referred to their respective clinics or our collaborating partner facility – Rakai Health Sciences Program (RHSP) to get tested, and to receive the appropriate treatment and care, as needed. 

Intro/Methods/results/Conclusions

4. STI prevalence instead of STIs prevalence

Response: We have addressed this throughout the manuscript.

5. What is the general population STI prevalence?

Response: Unfortunately, there are no national level data on the prevalence of bacterial STI in

the general population. Available data focus on specific subgroups, such as adolescents and

young women, WESW, etc. As such we are unable to provide these estimates. 

Also see: Kakaire, O., Byamugisha, J. K., Tumwesigye, N. M., & Gamzell-Danielsson, K. (2015).

 Prevalence and factors associated with sexually transmitted infections among HIV positive

 women opting for intrauterine contraception. PLoS ONE, 10(4), 1–12. 

6. STI prevalence is estimated at 13% for gonorrhea, 9% chlamydia, 10% syphilis and 17% for trichomonas [4]. Is this for genital STI’s?

Response: Yes

7. Please give the names of the tests you used for HIV and STI

Response: The tests have been added under “Measures: HIV and STI sample collection” as follows: Abbott Determine, Chembio statpak cassette and Abbott Bioline tests were used for HIV testing. NOVA was used to test for Neisseria gonorrhoeae, vaxpert for Chlamydia trachomatis and JD Biotech for Trichomonas vaginalis. 

8. Participants who were found to have STIs received single dose of treatment – rather say received targeted treatment for that STI.

Response: We have changed this statement to read as suggested. 

9. Age usually reported as a median

Response: Age was normally distributed, as such, we report the mean as it approximates the median.

10. STI levels seem lower than expected

Response: We acknowledge this comment. 

11. Report results consistently xx% (n= yy)

Response: We have addressed this throughout the manuscript. 

12. Women with more than one STI were 0.7%, - this does not make sense. Rather say xx women (0.7%) tested positive for more than one bacterial STI.

Response: We have revised this statement to read as follows: “Of the total sample, 4 (0.7%) women tested positive for more than one STI, and 32 (5.6%) tested positive for both HIV and STI.

13. STIs prevalence is estimated at 13% for gonorrhea, 9% chlamydia, 10% syphilis and 17% for trichomonas [4]. From intro. Same sentence in discussion says syphilis is 17% with same citation.

Response: We apologize for this oversight. We have corrected this statement to indicate the correct estimates for syphilis – at 10%.

14. Line 294 is obscured

Response: This was as a result of the pdf builder and has been corrected. 

15. The discussion makes some statements that seem to be more author opinion than actual study conclusions. The authors should refrain from hypothesizing reasons for associations unless these can be backed up by literature. For example saying that older women fear stigma more than young women could be completely incorrect. This is an author hypothesis and not a study result. You could say that xxx study showed this to be true and this could explain the association you found in your study. In the same way the authors should be very careful of suggesting interventions based on associations. It would be preferable to say “we noted an association between xxx and yyy and further research is needed to understand this association.

Response: We acknowledge these concerns and have revised these statements. For the first statement, we have revised as follows:

“Other studies suggest that older women are often not thought about when designing HIV prevention and reduction programs, and their concerns are rarely addressed by risk reduction interventions[45] . Moreover, low education has been documented as a risk factor for sexual risk behaviours [45] and exposure to HIV and other STI among key populations, including WESW[46,47]. “

For the second statement, we have revised as follows: 

“Study findings point to the need for WESW-focused efforts that address behavioural health risks to help reduce the burden of HIV and STI among this vulnerable group. Such efforts should emphasize access to user-friendly information about HIV and STIs risk reduction –taking into account their literacy levels. More economic empowerment programs addressing financial literacy gaps and access to startup capital may be important to help WESW focus on other sources of income, which may lead to better negotiating power with their customers regarding condom use. 

16. The discussion should also compare predictors of HIV in the general female population with WESW. It is understood that HIV prevalence is higher but would be interesting to understand if predictors are different.

Response: This has been considered and we have included predictors of HIV in the general female population under the discussion section. The predictors are more less the same for the general female population and WESW. 

17. Line 298 – Prevalence of HIV tends to increase with increasing age. The trend of increasing prevalence in HIV in WESW seems to be the same as for the general population and is an epidemiological phenomenon with a disease like HIV that is incurable.

Response: Thank you for your comment. 

18. Please provide evidence from your study supporting these statements: This may be attributed to stigma or fear of accessing preventive methods (e.g., buying condoms, PrEP and PEP services) by older women.

Response: This statement has been revised as follows: 

“Other studies suggest that older women are often not thought about when designing HIV prevention and reduction programs, and their concerns are rarely addressed by risk reduction interventions[45] . Moreover, low education has been documented as a risk factor for sexual risk behaviours [45] and exposure to HIV and other STI among key populations, including WESW[46,47]. 

19. Line 306 – please provide evidence for the following statement: Women from households with more children may be less likely to engage in high-risk behaviours for fear that children will notice. They may also be more likely to think about the continuity of providing for their children if they are the primary caregivers, therefore taking extra precautions/preventive measures about their lives compared to those with less children

Response: This explanation came from our community collaborative board members –including WESW. However, given the lack of documented literature, we have removed this statement from the manuscript. 

20. Line 306 should be “fewer” children. Do you have any evidence to back up your theories of why fewer children would be a risk factor for HIV. Could this be confounding?

Response: We have removed this statement from the manuscript. Same explanation as in #19 above. 

21. Line 330 needs to be rewritten

Response: We have revised this line to read as follows: “Women with low financial self-efficacy have less negotiating power with clients regarding safer sex, which puts them at high risk of HIV.”

22. Line 333 – Do you mean accepting attitudes rather than approving attitudes?

Response: Yes, this has been revised accordingly. 

23. Line 342 is incomplete

Response: We have revised this statement to read as follows: “Such efforts should emphasize access to user-friendly information about HIV and STIs risk reduction –taking into account their literacy levels. 

24. A major limitation of this study is potential confounding. The association between low levels of education and HIV may actually be caused by poverty. Educational programs may not be helpful in this context but poverty alleviation programs may work. The authors should be careful about speculating on interventions based on associations found.

Response: This has been considered and included in the limitations section

Reviewer #2: 

Although not the focus of this study, the authors did not explore the prevalence of HPV, which could have been an important opportunity for cervical cancer screening and prevention.

Response: We acknowledge this comment, and we hope to explore HPV prevalence in the future.

1. Were patients with a bacterial STI symptomatic?

Response: We acknowledge this comment. However, we are unable to respond as this information was not made available to us. 

2. Were patients with a detected STI treated and with what treatment?

Response: Yes, all participants with a detected STI were treated as follows: Azithromycin (1 gram) tablets for Neisseria gonorrhoeae and Chlamydia trachomatis, and metronidazole (2 grams) tablets for Trichomonas vaginalis. This information has been added under the “HIV and STI sample collection” section.

3. Line 139: could the author precise the apparatus and kit use for NAAT

Response: This information has been added under the “HIV and STI sample collection” section.

4. Line 222: The average age of participants was 31.4% in the text and 31.6% in the table 1. The status married was 25.7 in the text and 25.6 in the table 1

Response: We apologize for this error. We have corrected this, and the text is now consistent with the figures in Table 1. Specifically, the average age of participants was 31.6. About 87% (n=473) of the participants had primary education, and only 25.6% (n=139) were married or in a relationship. 

5. Title “Sample characteristics” change for “Description and characteristics of population studied “

- Legends of the Table: “Total Sample (N=542) % (n)” could be precise to understand that the author gives sometimes the %, or the number or the average score.

Response: This has been revised as requested. 

6. What means the number 7.18?

Response: This is standard deviation

---

## [Decision Letter · Decision Letter 1]

4 Aug 2022

Prevalence and Predictors of HIV and Sexually Transmitted Infections among Vulnerable Women Engaged In Sex Work: Findings from the Kyaterekera Project in Southern Uganda

PONE-D-21-34621R1

Dear Dr. Ssewamala,

We’re pleased to inform you that your manuscript has been judged scientifically suitable for publication and will be formally accepted for publication once it meets all outstanding technical requirements.

Kind regards,

Jianhong Zhou

Staff Editor

PLOS ONE

Additional Editor Comments (optional):

Reviewers' comments:

Reviewer's Responses to Questions

**Comments to the Author**

1. If the authors have adequately addressed your comments raised in a previous round of review and you feel that this manuscript is now acceptable for publication, you may indicate that here to bypass the “Comments to the Author” section, enter your conflict of interest statement in the “Confidential to Editor” section, and submit your "Accept" recommendation.

Reviewer #1: All comments have been addressed

2. Is the manuscript technically sound, and do the data support the conclusions?

Reviewer #1: Yes

3. Has the statistical analysis been performed appropriately and rigorously? 

Reviewer #1: Yes

4. Have the authors made all data underlying the findings in their manuscript fully available?

Reviewer #1: Yes

5. Is the manuscript presented in an intelligible fashion and written in standard English?

Reviewer #1: Yes

6. Review Comments to the Author

Reviewer #1: No further comments - all reviewer comments have been addressed

7. PLOS authors have the option to publish the peer review history of their article (what does this mean?). If published, this will include your full peer review and any attached files.

Reviewer #1: **Yes: **Katherine Gill

---

## [Editor Report · Acceptance letter]

31 Aug 2022

PONE-D-21-34621R1 

Prevalence and Predictors of HIV and Sexually Transmitted Infections among Vulnerable Women Engaged In Sex Work: Findings from the Kyaterekera Project in Southern Uganda 

Dear Dr. Ssewamala:

I'm pleased to inform you that your manuscript has been deemed suitable for publication in PLOS ONE. Congratulations! Your manuscript is now with our production department. 

Kind regards, 

on behalf of

Jianhong Zhou 

Staff Editor

PLOS ONE